# Two Decades of Architects' and Urban Planners' Contribution to Urban Agriculture and Health Research in Africa

**Akuto Akpedze Konou** [1,*] , **Armel Firmin Kemajou Mbianda** [1], **Baraka Jean-Claude Munyaka** [1]
**and Jérôme Chenal** [1,2]

1   Communauté d'Études Pour l'Aménagement du Territoire (CEAT/EPFL), École Polytechnique Fédérale de Lausanne (EPFL), 1015 Lausanne, Switzerland; armel.kemajou@epfl.ch (A.F.K.M.); baraka.munyaka@epfl.ch (B.J.-C.M.); jerome.chenal@epfl.ch (J.C.)

2   Center of Urban Systems, Mohammed VI Polytechnic University, No. 660, Hay Moulay Rachid, Ben Guerir 43150, Morocco

*   Correspondence: akuto.konou@epfl.ch; Tel.: +41-21-693-77-10

**Abstract:** Urban agriculture (UA) is an ancient practice in Africa that meets social- and health-related needs. However, it is unclear whether architects and urban planners have incorporated the topic into their research and practices. This study aimed to assess the scientific contributions of these fields to UA and their relevance compared to other disciplines. The research objectives were to evaluate the trends in the subject, architects' and planners' involvement, and the effects of UA on health in Africa. As a method, a review was conducted using Scopus, PubMed, Web of Science, and Google Scholar. The research query was ("urban agriculture" OR "urban farming") AND Africa AND ("health" OR "global health" OR "urban health"), and the time frame considered was January 2000 to December 2020. Zotero, Mozilla Firefox, Google Chrome, Excel, and VOSviewer were used to collect and analyze metadata. After excluding duplicates, a total of *n* = 390 articles were involved. The results displayed the mixed health effects of UA, a growing interest in the topic with prominence on food security, and evidence from public health, not architecture and planning. The study recommends more theoretical research on UA by architects, which should be translated into policies and implementation.

**Keywords:** land use policy; literature review; urban agriculture; urban health; urban planning

## 1. Introduction

### 1.1. Historical Background

Urban farming (UF), or urban agriculture or (UA), describes the practice of agriculture within urban and peri-urban settings [1]. UA is an ancient practice, especially on the African continent. For instance, Juhé-Beaulaton's [2] historical investigations in Dahomey (Benin) demonstrated that plant cultivation in the cities of the Gulf of Guinea began prior to the arrival of European travelers. The author drew on the accounts of travelers and recorded oral testimonies. According to Sibhatu and Qaim [3], farming contributed to the daily subsistence of these people in Africa, impacting most strata of society, even in urban areas. Indeed, the practice of UA has persisted in Africa throughout and after the colonial period in order to "meet the consumption needs of bureaucrats, colonists and other elites" [4]. In the post-independence period, and the intense rural exodus that followed, former inhabitants of rural areas who found themselves in cities and without jobs started practicing UA because they needed an income-generating activity to live and survive in the city [5].

### 1.2. Contemporary Relevance

Among other capacities, a city must have the means to feed its inhabitants [6]. In addition to other "vital" functions, guaranteeing food security is an essential concern, with particular relevance during the COVID-19 pandemic [7]. In fact, UA has become a key

contributor to food security in cities [8–10] in addition to employment and sustainability, notably in Africa [11,12]. These issues have become most pressing in developing regions due to the intensity of their urban growth. According to Nagendra and colleagues [13], "Ninety per cent of the projected world population growth of 2.5 billion over the next couple of decades will occur in the cities of Africa and Asia".

*1.3. Problematic and Gaps*

Despite their historical synergy, the fields of urban planning and public health have been progressively disconnected [14]. In this regard, UA might be a fruitful platform to use in order to reconnect those disciplinary fields [15]. For instance, recent research has raised concerns about urbanization's impact on ecosystems and the potential for UA to mitigate some of these effects [16]. From a broader perspective, UA can be approached from social, political, and environmental perspectives, providing insights into its significance to cities in Africa and on other continents [17]. Moreover, as Africa experiences unprecedented urban growth [13], integrating specific urban policies to address UA (which is already an ancient practice on the continent) could help to mitigate the negative effects of such rapid urban growth.

This study addressed a knowledge gap regarding the recent contribution of urban planners to scientific discussions on UA in Africa. It explored the extent to which urban planning scholars have addressed the complex relationship between UA and health in African cities. It utilized a broad, interdisciplinary approach, with an emphasis on the potential role of urban planners in fostering adequate spatial frameworks that allow UA to contribute to urban health.

Why is it timely to explore such a study?

Studying the relationship between UA, planning, and health is timely for several reasons, including the context of rapid urbanization, health challenges, environmental sustainability, policy and planning, and equity and social justice.

What makes this study different from other studies in the world?

This study differs from other studies worldwide in several unique aspects and contributions, considering points such as the focus on Africa, role of architects and urban planners, length of analysis, overview, interdisciplinary approach, implications for African urban policies, and influence of actors.

Moreover, the literature review shows that the study differs from prior studies and that there are no publication studies that focus precisely on the same subjects within the same scope.

*1.4. The Research Objectives*

This literature review focused on how UA and urban health in Africa have been addressed by researchers in architecture and urban planning in relation to other fields. What motivated this investigation was the need for architects and urban planners to scrutinize the characteristics of and interconnections between UA and urban health. This research aimed to provide a comprehensive appraisal of the favorable and unfavorable impacts of UA on public health in African cities, thus emphasizing potential implications for urban policymaking across Africa.

The underlying hypothesis was that, compared to other fields, architecture and urban planning researchers have not substantially contributed to the scientific discussion on UA, even though they could play an important role in reducing its health risks while increasing its benefits through adequate spatial and regulatory frameworks. To test this hypothesis, three specific objectives were derived:

- The first was to assess the general evolution of interest in UA–urban health linkages in recent decades (2000–2020);
- The second was to assess the specific role of urban planners in the knowledge production on UA and health in Africa by quantifying the number of works in this field as compared to other fields;

- The third and last objective was to establish an overview of the positive and negative health impacts of UA in African cities.

## 2. Materials and Methods

### 2.1. Data Collection

The databases used to conduct the review were: Scopus, PubMed, Web of Science, and Google Scholar. Figure 1 is a schematic representation of the steps taken to collect the data used in this study. Other criteria included the word "health" and the concepts "global health" and "urban health". Finally, the labels "urban agriculture", "urban farming", "africa", "health", "global health", and "urban health" were adopted. The research query equation was defined after several trials in order to obtain as many accurate documents that match the key words as possible. The following query string was formed: ("urban agriculture" OR "urban farming") AND Africa AND ("health" OR "global health" OR "urban health").

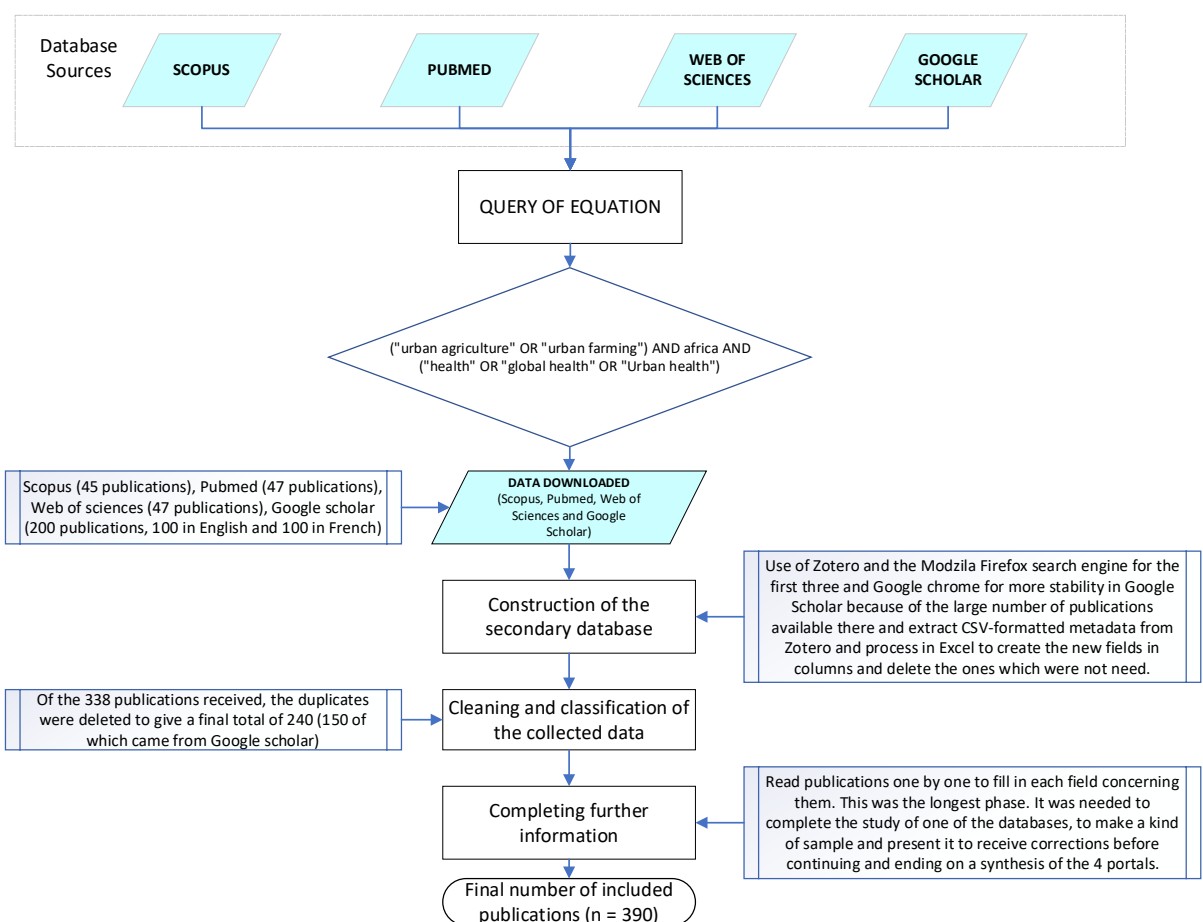

**Figure 1.** Data collection phase.

Experience has demonstrated that the results obtained can vary depending on the choice of web browser. This variability is particularly noticeable when using Google Scholar. For example, the royalty-free literature collection software Zotero 5.0.84 was used, deploying the search engine Mozilla Firefox for the first three. Additionally, Google Chrome was utilized on account of the greater result stability offered by Google Scholar due to the large number of publications available.

For an in-depth analysis of the data, the metadata were extracted in CSV format from Zotero. They were then processed in Microsoft Excel to present the new fields in column

form and delete unnecessary fields. Finally, in the most tedious stage, all publications were read one by one to fill in each field.

The data cleaning process used was the one advised by Rahm et al. in 2000. The method has five phases: (i) data analysis, (ii) definition of transformation workflow and mapping rule, (iii) verification, (iv) transformation, and (v) backflow of cleaned data [18], as stated in [19]. The main difficulty was reading each piece of data word by word and line by line. This was tiring as it was performed manually.

*2.2. Analysis*

2.2.1. Analysis Using Microsoft Excel 16

Once the data were collected, cleaned, and arranged, Microsoft Excel 16 was used to produce graphs. Additionally, QGIS 3.32.2 produced maps showing the geographic distribution of the included works. Analyses were conducted as follows:

(1) A typological analysis: Since this research included both gray and scientific publications, a typological profile of these publications was established. The metadata in Zotero's "Item Type" column show whether the publication was a book, a thesis, a journal article, a conference paper, or a book section. The results of this analysis provided insights into the preferred publication types within each field;

(2) A numerical analysis: A systematic count was performed for all publications from the year 2000 to 2020 to evaluate the chronological evolution. The goal was to determine if the subject has been of interest or not during the last 20 years;

(3) A departmental/institutional analysis: An analysis of the different disciplines from which the publications come was undertaken. These affiliations were found by going back to each online article, clicking on the first author of each publication, and seeing the institutions in which they worked while writing the paper. This analysis allowed us to define the fields involved in researching the subject and to see whether urban planners made a significant contribution during the reference year;

(4) A thematic analysis: Under the overall umbrella of UA and health, more specific studies were conducted. A thematic analysis served as the appropriate method for assessing these diverse subjects. Based on the list of titles, abstracts, and keywords, the dominant topics were identified. These, in turn, were classified under themes by covering them with a group of globalizing terms. This phase of the analysis estimated the themes most commonly addressed in order to understand their importance in the field of research and to evaluate whether the research orientation according to these themes is driven by reality;

(5) A chronological evolution of topics: In addition to defining the types of topics that were dealt with between the years 2000 and 2020 under the broad theme of UA and health, a chronological schematization was conducted. The latter served to analyze the movement of these topics throughout the reference period;

(6) A geographical analysis of the study sites: This consisted of identifying the countries concerned by the research. Publication titles, keywords, and abstracts were used to find the country or countries chosen as case studies. A table was then drawn up that detailed these countries and the number of publications per country;

(7) A geographical analysis of the affiliation of the first authors: the second geographical analysis was about the countries in which the first authors were based;

(8) A general health impact assessment: This consisted of determining whether the selected publications considered UA to be more of a health risk than a benefit or to represent aspects of both. This was defined based on the title, abstract, and conclusion.

2.2.2. Analysis Using VOSviewer

To enhance the depth of our study, we also employed a bibliometric analysis using the VOSviewer to extract publications (from 2000 to 2020) from three databases: PubMed, Web of Science, and Scopus. The VOSviewer does not support Google Scholar.

Searches were conducted in all three databases using the same query previously mentioned, covering 1 January 2000 to 31 December 2020. The information collected included citations, bibliographies, abstract and keywords, and funding details. We gathered 67 documents from Scopus, 74 from Web of Science, and 94 from PubMed, resulting in 235 publications (Table 1).

**Table 1.** Settings for abstract analysis with VOSviewer.

| Settings | Web of Science | PubMed | Scopus |
|---|---|---|---|
| Type of data | Create map based on text data | | |
| Data source | Read data from bibliographic database files | | |
| Fields | Abstracts | | |
| Counting method | Binary counting | | |
| Threshold (2) | 461 (of 2797) | 317 (of 2110) | 297 (of 1825) |
| Number of terms to be selected (60%) | 277 | 190 | 178 |

The VOSviewer extracted noun phrases from publication abstracts in the three aforementioned databases, generating output files that displayed a map of nodes or circles, each representing a noun phrase. The map employed colors to denote clusters, and lines connected the nodes, considering the proximity of the words. Larger nodes corresponded to nouns with higher frequencies of occurrence. Typically, when two nouns were positioned closely in the visualization, they signified a stronger bibliographic coupling between them. The study used colors to indicate clusters of researchers who exhibited relatively strong connections with one another.

The study subsequently imported data extracted from the three databases into VOSviewer for analysis and visualization. The study opted for circles to optimize visualization, allowing us to use concise labels of up to 30 characters. The default limit of 1000 lines was maintained for maximum visibility, and curved and colored lines were used to enhance the visual representation.

The study conducted two distinct types of analysis for each database. The analysis was based on the content of publication abstracts. The table below provides details of the specific parameter choices made for these analyses.

### 3. Results

*3.1. Results Using Microsoft Excel 16*

By database, here is the number of publications that were obtained:

Scopus ($n$ = 45 publications), PubMed ($n$ = 47 publications), Web of Science ($n$ = 47 publications), and Google Scholar ($n$ = 150 publications).

#### 3.1.1. Types of Publications

The review of the types of publications around this general theme picked among the academic peer-reviewed articles and the gray literature showed that there was a more significant number of journal articles (83%) than books (7%) or conference papers (6%). There was no book section at all in the PubMed list.

#### 3.1.2. Variation in the Number of Publications over the Last 20 Years

From the analysis, the number of publications over all 20 years (Figure 2) reveals an average growth, peaking in the 2006–2010 quintile (precisely $n$ = 24 publications in 2010). A drop in value by half followed in 2011, a state that persisted until 2017. However, since then, the number of papers on the subject seems to have grown, and 2018–2019 in particular saw growth (respectively, $n$ = 21 and $n$ = 20 publications).

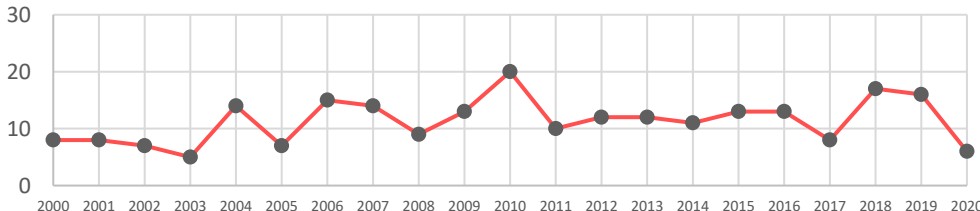

**Figure 2.** Number of publications per year over the last 20 years.

3.1.3. Main Fields of Affiliation of First Authors in a Logarithmic Scale (Only with Scopus, Web of Science, and PubMed Data)

Among the fields or types of departments that have already initiated research under the general theme (Figure 3), the most prevalent are public health (*n* = 34 publications, or 21%), earth and environmental science (*n* = 18 publications, or 11%) and agriculture (*n* = 16 publications, or 10%). Zoology trails in popularity, chased by water management.

**Figure 3.** Fields of study.

### 3.1.4. Areas of Research

The most widely encountered themes are food security (22%), wastewater irrigation risks (12%), broad characteristics of UA as experienced on the African continent (11%), malaria risks (8%), and the health impacts of UA considered in general (7%). The socio-economic status and benefits topic is less commonly discussed than the land use subject (5% each). At the end, an image is obtained, with the words that appeared the most in the list appearing with a larger size, a more central position, and a greater writing thickness. In the following images from monkeylearn.com (accessed on 21 September 2023) (Figure 4), the words "food security" and "human", "malaria", and "wastewater" are the most prominent other than "urban" and "urban agriculture", which are already keywords in the search equation. The Monkeylearn online word cloud generator was selected for this paper due to its capability to highlight the 10 most pertinent words based on frequency, thereby enhancing reader comprehension.

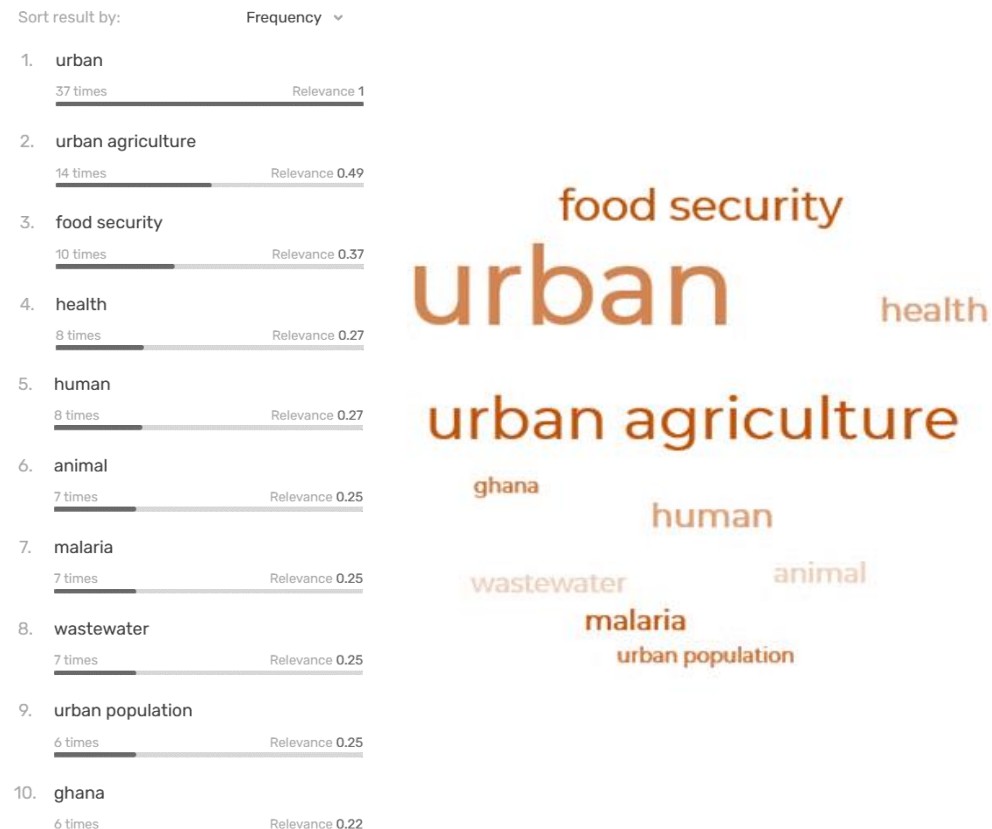

**Figure 4.** Publication topics word cloud generated via monkeylearn.com (accessed on 21 September 2023).

### 3.1.5. Chronological Evolution of Topics (Only Scopus)

A chronological study was performed to produce a representative graph of the evolution of the topics dealt with over the years (Figure 5). To categorize the studied publications, they were classified by general themes based on the main topics of their contents. Thus, 22 subtopics have been elaborated and classified by year and by number of appearances. The themes are as follows: city sustainability benefits, wastewater irrigation risks, policies, zoonoses, UA general, the literature review of UA pros and cons, green infrastructure support, general impacts on health, malaria risks, socio-economic status in UA, land use, UA characteristics, pesticide/insecticide contamination dangers (surface waters), the literature review, waste management support, town refuse ash risks, ecology, urban floods fighter, metal sediment pollution (surface waters), eco-friendly pesticide benefits, and wastewater

irrigation benefits. The following table (Table 2) provides context or an explanation for each theme or its significance.

**Table 2.** General themes and significance (only Scopus).

| Number | Global Themes | Explanation |
|:---:|:---:|:---|
| 1 | Cities' Sustainability Benefits | Publications talking about the social, economic, and environmental benefits of UA. These texts often clearly include the term "sustainability". |
| 2 | Wastewater Irrigation Risks | Topics concerning irrigation as a method of supplying water to plants in urban agricultural fields. |
| 3 | Policies | Texts that deal with the need to introduce the subject into policies. |
| 4 | Zoonoses | Texts that talk about the diseases created by the animals that the UA attracts. |
| 5 | UA General | Texts that talk generally about the subject, touching on a bit of everything. They often express general points of view on the subject. |
| 6 | Literature Review of UA Pros and Cons | Literature reviews that deal with the pros and cons of UA for health. These are not negligible. |
| 7 | Green Infrastructure Support | Publications that argue that UA is a priority "green infrastructure" for the city. |
| 8 | General Impacts on Health | Publications that talk generally about the subject, without focusing on something. |
| 9 | Malaria Risks | Texts that find that UA contributes to the attraction of mosquitoes. |
| 10 | Socio-economic Status in UA | UA as a positive contribution to the socio-economic status of users. |
| 11 | Land Use | Texts that deal with the lack or importance of integrating UA by zoning in urban planning. |
| 12 | UA Characteristics | Publications that describe the parameters of urban agriculture, detailing the types and methodologies of their practice. |
| 13 | Pesticide/insecticide Contamination Dangers (Surface Waters) | The texts consider UA to be a health hazard because of the pesticides/insecticides farmers use, which can be toxic to the produce. |
| 15 | Literature Review | Texts that have performed a general literature review of AU to see what results they come up with without specifying anything else. |
| 16 | Waste Management Support | Publications that argue that UA practice is very much part of Waste Management Support. |
| 17 | Town Refuse Ash Risks | Texts that focus on the use of Town Refuse Ash Risks as inputs into UA. |
| 18 | Ecology | Publications that support UA as a priority contribution to ecology. |
| 19 | Urban Floods Fighter | Publications that support UA as a means of combating flooding in parts of cities by protecting the land. |
| 20 | Metal Sediment Pollution (Surface waters) | Texts that focus on the contamination of UA products via metal sediments. |
| 21 | Eco-friendly Pesticide Benefits | Publications that talk about the use of organic pesticides and their benefits. |
| 22 | Wastewater Irrigation Benefits | Publications that talk about Wastewater Irrigation in UA and its benefits. |

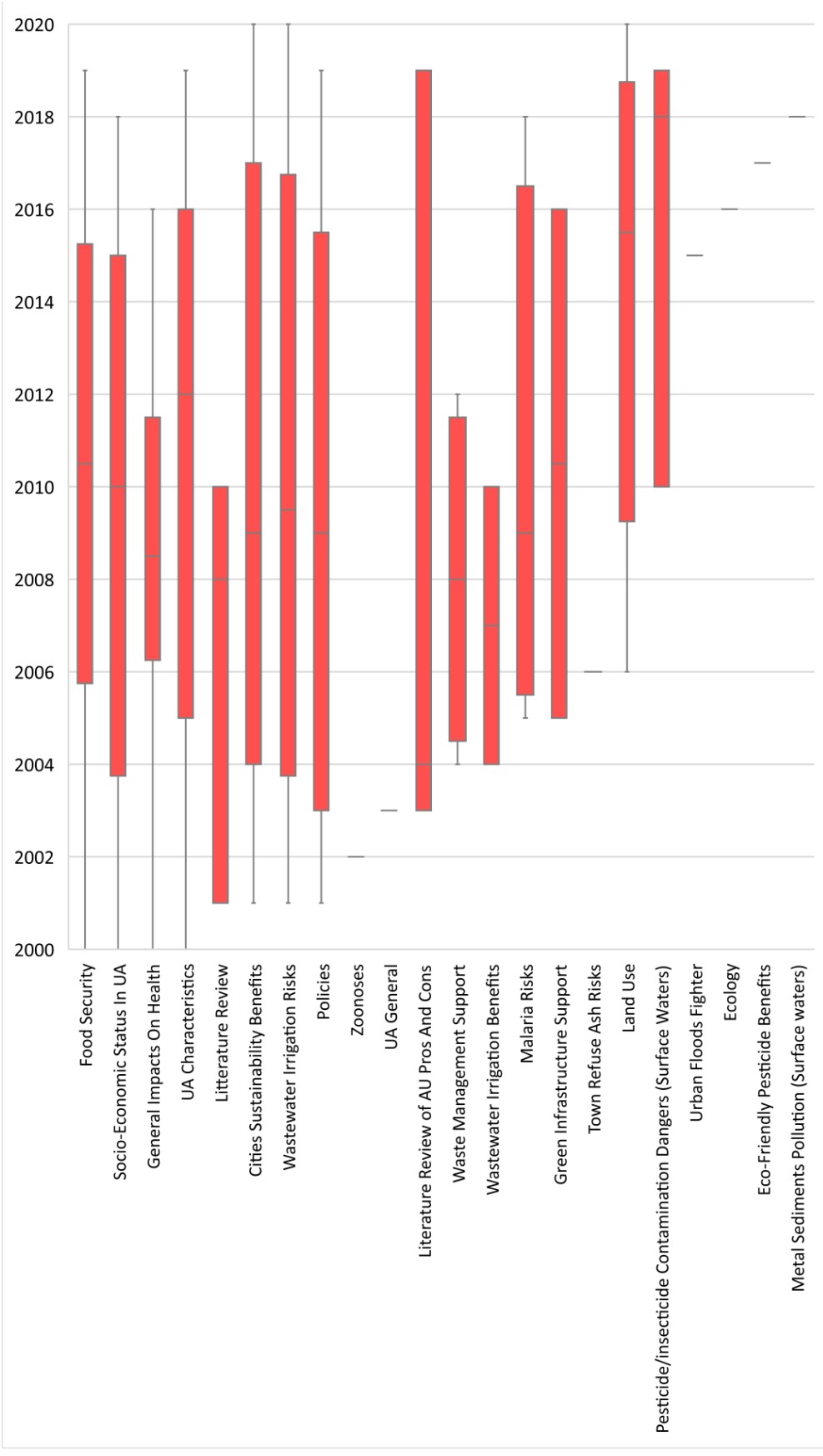

**Figure 5.** Chronological evolution of topics (only Scopus).

These specific topics were picked up from the databases because they globalize other sub-themes within themselves, making it easy to explain the overall themes at play. They are relevant to the overall research in the sense that they synthesize several small themes addressed by the authors that, if all listed, would be overwhelming for readers of this paper to grasp. The following graph shows the behavior of these UA trends throughout the years from 2000 to 2020.

In Figure 5, the black bars represent the number of studies published per field by year. The standard deviation (-) shows the measure of how dispersed the rate of publication per discipline per year is in relation to the mean. This graph shows that among the 22 topics that emerged, "food security", city sustainability benefits, wastewater, and "literature review of UA pros and cons" are those that have been the most consistent, having been addressed at least every year from 2000 to 2020.

As much as with the topic "literature review of UA pros and cons", publications concerning "pesticide/insecticide contamination dangers (surface waters)" have gained extensive popularity in recent years and since the 2010s. The subject of "land use" has also seen some growth, almost paralleling the theme of "pesticide/insecticide contamination dangers (surface waters)".

The topic of wastewater irrigation benefits was found to be the least discussed theme over time.

The topics of metal sediment pollution, waste management, and general impacts on health and food security support have specifically gained interest in the last five years.

The questions of "zoonoses", "UA general", and "town refuse ash risks" are the topics that have been least commonly handled during the period covered by this study of the literature review.

### 3.1.6. Localization of the Studied Cases (Only with Scopus, Web of Science, and PubMed Data)

The case studies for the Scopus portal mainly come from Ghana (14%), Côte d'Ivoire (8%), and South Africa (8%). Some publications dealt with regions rather than countries, but not all regions have official boundaries distinct from one another. These data have not been considered. These are "Africa" ($n = 5$), "Equatorial Africa" ($n = 1$), "Global South" ($n = 1$), "Intertropical Africa" ($n = 1$), "Sub-Saharan Africa" ($n = 3$), and "West Africa" ($n = 2$). A map shows two major regions of concentration: West Africa and East and South Africa.

The Maghreb countries and Madagascar are hardly represented in the accounts. Ghana, Côte d'Ivoire, and Burkina Faso are three bordering countries with non-negligible numbers of publications on the subject. The same is true for Tanzania, Kenya, and Uganda.

It is also noted that among the more frequently represented countries, most have larger land areas, such as South Africa, Tanzania, Kenya, Nigeria, and Cameroon. Referring to the map obtained after data visualization using QGIS 3.32.2 (Figure 6), coastal countries also seem to have made a special effort to study the subject.

### 3.1.7. Affiliation of Authors

Of the $n = 115$ publications collected and studied on Scopus, PubMed, and Web of Science, less than 20% were initiated by the laboratories of affiliation of the first author located in Africa. This fact is shown in Figure 7, obtained using QGIS 3.32.2. Thus, the UK, the USA, and Switzerland (perhaps because of the IP address located in Switzerland) are the countries that have dealt with the subject the most, with rates of 16.1%, 14.95%, and 14.95%, respectively. Even if they are not in the winning trio, South Africa and Ghana each comprise 10% of the sum of the publications.

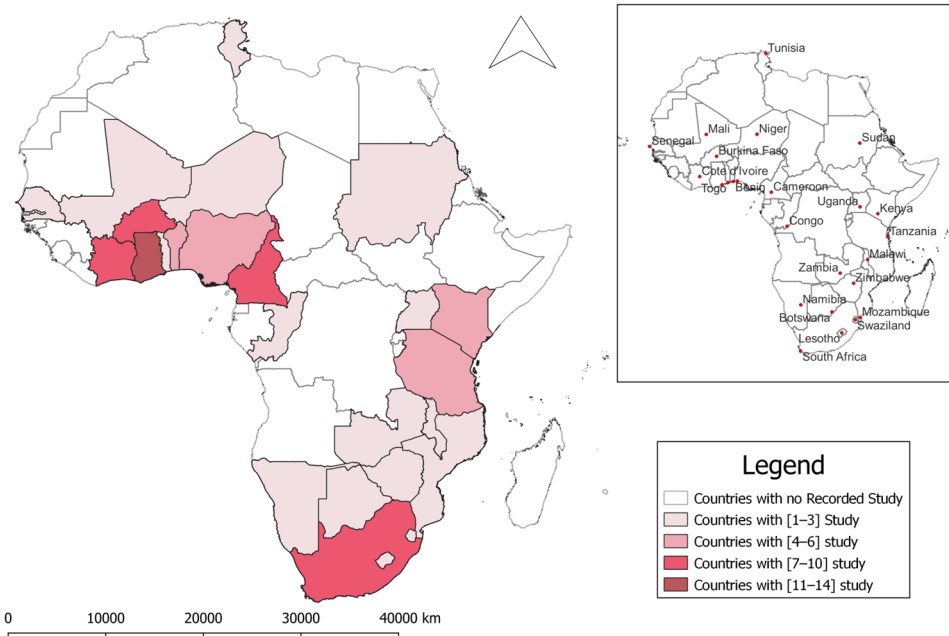

**Figure 6.** Localization of the studied cases from Scopus, Web of Science, and PubMed.

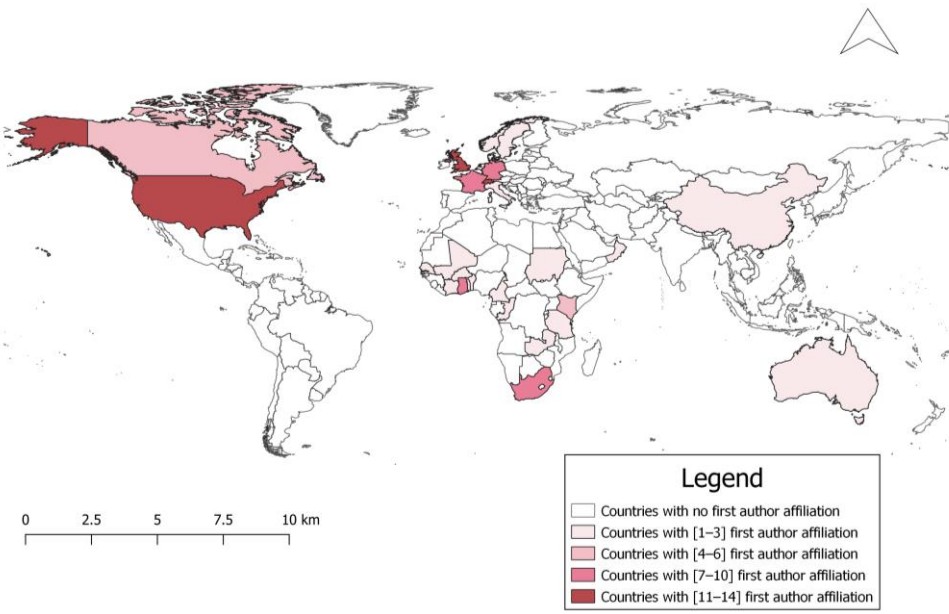

**Figure 7.** Countries with affiliation of authors.

3.1.8. Does UA Have a Negative or a Positive Impact on Health?

Finally, in this study, "positive impact" is defined as a favorable effect or outcome of the practice of UA on the health of either the farmers or consumers of these crops. This definition contrasts with "negative impact", an effect or outcome of UA that is detrimental to health. The third category, called "positives and risks", is considered to encompass publications that have hypothesized both positive and negative health effects of UA.

In this research, it was assessed how the results of this research about UA were viewed and the impacts of the practice on health in Africa (Figure 8). About 48% of the results speak of positive impacts, but they are accompanied by risks to be managed. Some 29% found that UA has totally positive effects on the health of urban dwellers, and 23% highlighted the negative effects of this practice in African cities.

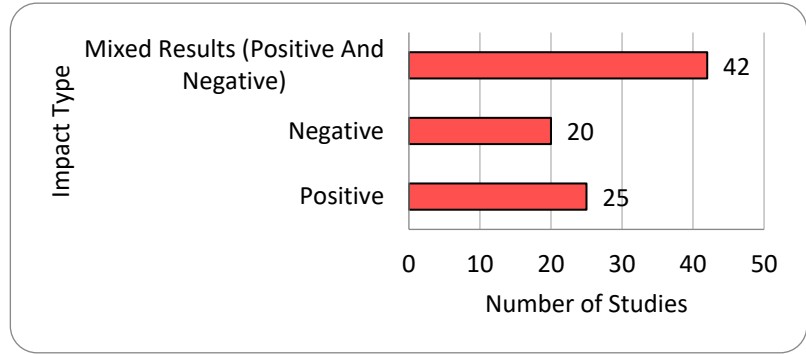

**Figure 8.** Number of studies per type of impact.

*3.2. Results Using VOSviewer*

3.2.1. Results from Web of Science

Figure 9 depicts the visualization of the term co-occurrence network within Web of Science abstracts. The VOSviewer has organized these terms into eight clusters, with five of them being notably large. These five scientometrics encompass terms such as "irrigation water", "season", "insecticide", "survey", and "malaria". The clusters shaded in yellow-green and orange, located on the left side, pertain to public health, while the green and red clusters in the lower left corner are associated with policies. Moving to the upper part of the visualization, the violet and sky-blue clusters are composed of terms linked to countries and cities. In contrast, the darker blue cluster in the upper left mainly focuses on food-related terms. Notably, the network connections centered around "insecticide", "season", and "survey" are the most prominent.

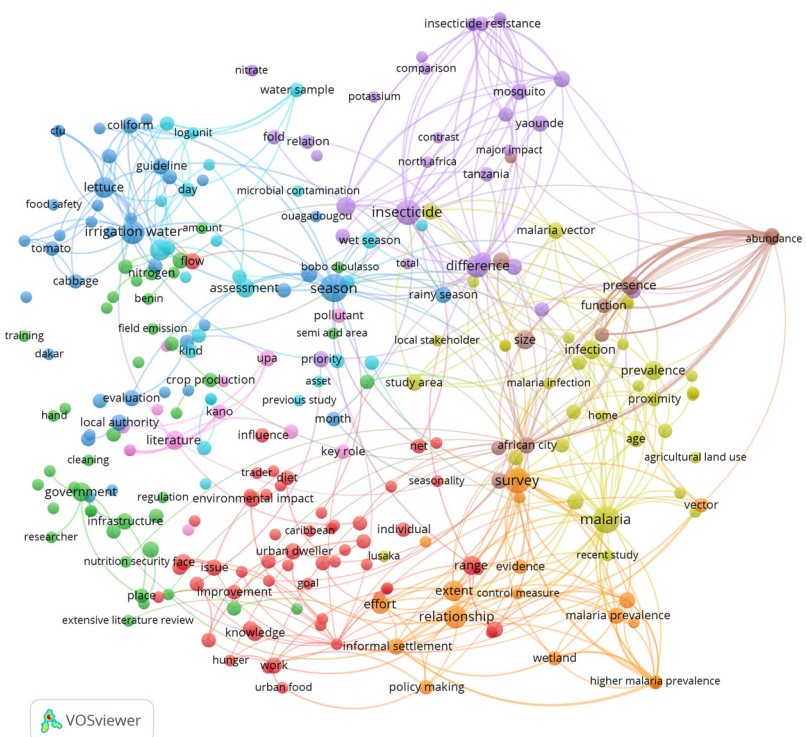

**Figure 9.** Web Of Science abstracts' visualization term co-occurrence network.

3.2.2. Results from PubMed

In Figure 10, the visualization of the term co-occurrence network within the PubMed database's abstracts is presented. The software has grouped these terms into six clusters, with four of them being notably substantial: "survey", "irrigation water", "season",

"Ghana", and "malaria". The green and blue clusters on the right side are associated with public health and location, and they are surrounded by numerous connection links, indicating their strong relationships within the network.

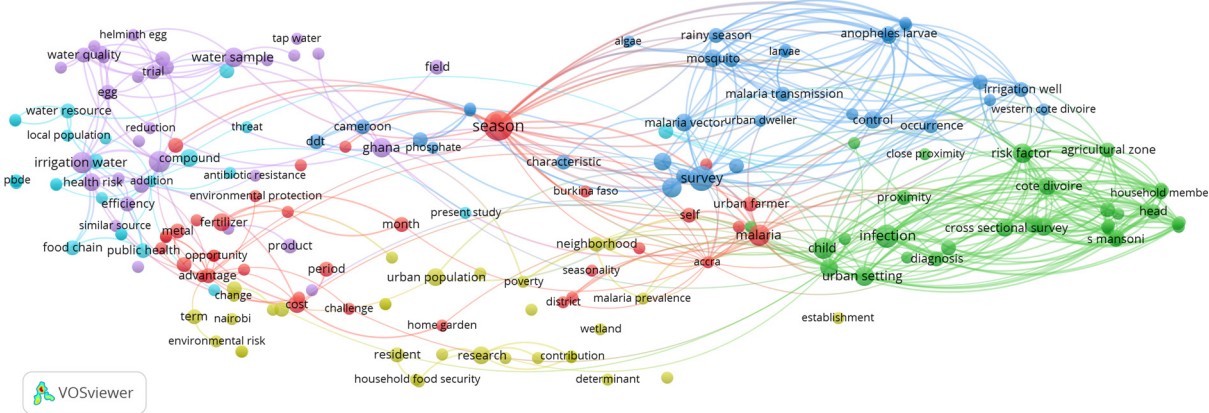

**Figure 10.** PubMed abstracts' visualization term co-occurrence network.

### 3.2.3. Results from Scopus

In Figure 11, the emergence of five clusters characterized by prominent terms such as "access", "prevalence", "site", "malaria", "Ghana", "infection", and "insecticide" is observed. The topmost violet cluster pertains to geographical locations, while the green cluster positioned in the lower-right corner illustrates the operational practices of the AU. A notable interaction occurs among these regions and the lower-left sector, encompassing two additional policy-related clusters. The yellow cluster connects these three regions, featuring pivotal terms like "site" and "insecticide".

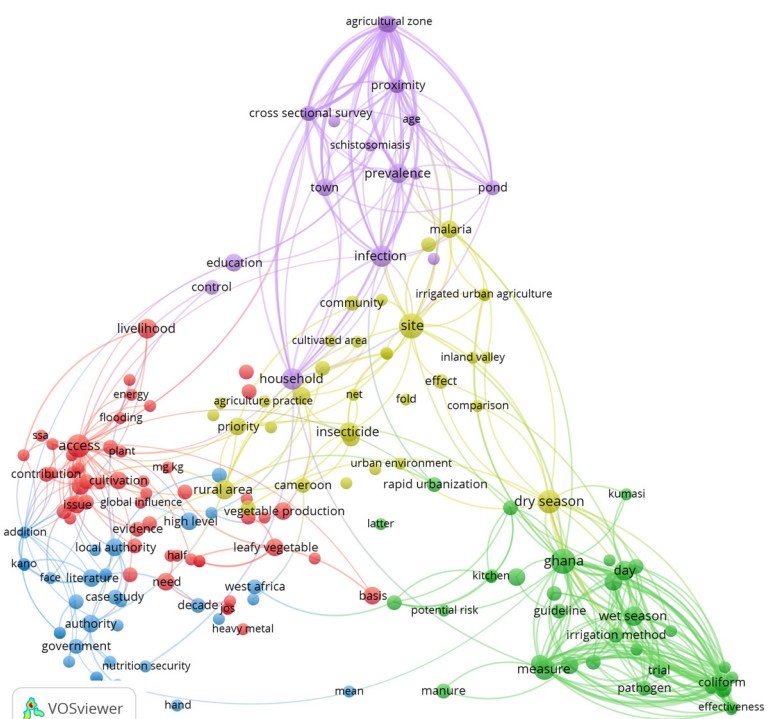

**Figure 11.** Scopus abstracts' visualization term co-occurrence network.

In conclusion, Figures 9–11 visually represent term co-occurrence networks within abstracts from Web of Science, PubMed, and an AU dataset, respectively. In Figure 9, VOSviewer identifies eight clusters, with five significant ones, including terms like "irriga-

tion water", "season", and "insecticide". These clusters relate to public health, policies, and more. In Figure 10, the PubMed dataset yields six clusters, notably "survey", "irrigation water", and "Ghana", emphasizing the connections between public health and location. Figure 11 reveals five clusters, with terms like "access" and "malaria", showcasing connections among geographical locations, UA practices, and policy themes, bridged by a yellow cluster featuring "site" and "insecticide".

## 4. Discussion

In this review of the last two decades of publications on UA and health in Africa, studying the Scopus, PubMed, Web of Science, and Google Scholar databases, the analysis showed a growing interest in the research world and revealed the lack of research conducted under the auspices of urban planning for this topic. The following lines put the results found above into perspective in terms of specific objectives, recommendations, and limitations.

### 4.1. Assessment of the General Evolution of the Interest in the UA–Urban Health Linkages in Recent Decades (2000–2020)

#### 4.1.1. The First Place of Scientific Articles in Systematic Literature Publications

The results of this part of the analysis legitimize the notion that more master's and PhD theses, reports, books, and conference papers on this subject would be welcome. Indeed, even though research allowed itself to cast a wide net in even the gray literature, articles are preponderant over other types of publications on the subject, as is often the case [20].

Furthermore, the beneficiaries of the study must be precisely defined [21]. Objectively, it would be primarily the urban producers and consumers of UA products in these African cities and experts in urban planning, agriculture, and health. One possible explanation could be that databases receive far more articles for publication than any other type. Another could be, for example, that articles take fewer research resources than books, while books have an economic model that requires them to go through specific publishing houses.

#### 4.1.2. UA and Health Topics Gain Ground

Considering the variation in the number of publications over the last 20 years, it was noticed that the subject is gaining interest, indicating that the research presented here fits well into this dynamic. The result emphasizes the objective of the urge for architects and urban planners to join the movement. This fact could be justified by the increase in ecological objectives on the agenda of international institutions, causing a rise in funding for research projects related to UA and organic and environmental production [22].

Also, the importance of greenery in general in cities, given its myriad of benefits, as well as UA and its increasingly popular innovations, could contribute to the growth of interest in the subject [23]. Conversely, the risks posed by the practice of this activity in African cities, which make it controversial, could have led researchers and politicians to warn all stakeholders against this activity and the institutions on which it may depend [24].

Another reason could be that the issue of food security is becoming more and more prominent in the literature, even among SDG research. Many researchers may find themselves with more incentives to undertake research in this area.

#### 4.1.3. Many African Countries Are Waiting to Be Studied

Countries such as Equatorial Guinea, Sao Tome and Principe, Namibia, Botswana, Mauritania, and Angola saw a significant increase in urban population between 2000 and 2020. However, the analysis indicated that these countries have not produced studies at all regarding UA and health in this period.

Could this result be linked to the size of these countries, which do not have enough land for UA? Or could it be because they do not practice UA at all? Another possibility is that these countries may be applying modern urbanism extensively, thinking that UA is like bringing the village back to the city. However, the more diverse the countries studied

are, the more credible the results will be due to the heterogeneity and robustness of the samples [25–27].

### 4.1.4. The Imbalance between the Origin of the Research Laboratories

In terms of the results obtained, a mixed picture emerges regarding the number of publications from Africa. The case could be explained by the limited resources invested in research in the Global South. However, the hypothesis is that Africa would benefit from publishing much more in the areas of UA and health studies in Africa. On one level, the studies would be even more relatable because they would have been developed by people immersed in the realities of the context [28–30]. On another level, these publications could highlight innovations of which the Western world is not yet aware, confirming the concept of reverse innovation, which would consist of a South–North movement of "low-tech cutting-edge" methods, i.e., techniques that do not require a great deal of material resources but that solve real problems using technology [28,29].

Taking the specific cases of Tanzania, a predominantly English-speaking country in East Africa, and Togo, a negligible French-speaking country in West Africa, Tanzania is much better off than Togo as a field of study in this research area. This could be explained by local communities already supporting substantial UA governance in Tanzania [30]. Also, there is a growing number of international and civil institutions promoting and implementing UA planning efforts in Tanzania [31–34], while this topic remains embryonic in Togo.

### 4.1.5. Mixed and Qualitative Methods Are Underrepresented

The results achieved suggest that applying qualitative or mixed methods would contribute to the field of research under this theme. The superiority of the quantitative method may be due to the fact that agriculture is viewed through this lens from a more health-related perspective. This fact is evidenced by the fact that the two primary research disciplines are public health, earth, and environmental sciences [35]. This may have directed research towards the use of more quantitative methods.

However, of the two approaches used to improve under this theme, the mixed method can be particularly positive as, when translated into case studies [36], it can show a broader view of an issue at the case study level [37]. The intersection of the quantitative and qualitative study results allows for the harmonization and solidification of the results and, therefore, of the recommendations that could be drawn from them [38].

### 4.1.6. Is "Food Security" the First Objective for Which UA Should Be Practiced?

Considering all that the analysis of the topics has revealed, it would be fascinating to delve into the real aspects of food security, a limited goal that UA can help to achieve in African countries [39], to confirm or invalidate the objective since research related to food security may appear to be obvious [10].

Suppose the planner's point of view is to be considered. In that case, several theories are already confirmed about food security, with verification methods proving to be reasonably accessible to the planner, such as chronic disease primary and secondary data analysis using ethical methods [40]. Still, it is probably fair to say that the most recent policy analysis comes from agricultural circles rather than urban planning sectors [41].

Another way is to determine whether other, more subtle health outcomes are irrelevant in African cities, as they are in Europe and the US [42]. For example, is psychological well-being related to the socio-economic impacts of UA on urban health worthy of analysis? If so, is it equally distributed between genders among African urban farmers [43,44]?

*4.2. Assessment of the Specific Role of Urban Planners in the Knowledge Production on UA and Health in Africa by Quantifying the Number of Works in This Field as Compared to Other Fields: Urban Planning, One of the Poor Relations in the Literature on the Subject*

Regarding this paper, the information gathered through the review analysis shows that there needs to be more research at the level of the urban planning field.

Is this lack of publication in the field of urban planning justified because there are fewer publications in general by architects, urban planners, or any other urban planning experts who conduct more "practice-based research", as has been complained about [45]? Or is it because agriculture is generally considered a rural activity [46]? Is it because research shows that it is a phenomenon that has always existed in African cities and is even expanding [47,48] and that it does not require that much attention?

Another thing that could support this matter is that architects and urban planners might be more interested in the operational side of their profession for the immediate benefits that this can bring. In summary, one of the significant contributions of this article is not only its revelation of the growing importance of UA and its significant impact on public health, but also its emphasis on the need for architecture and urban planning disciplines on the African continent to take a greater and more concrete interest in this subject. This presents an additional empirical reason for integrating UA into their studies, planning documents, and development projects. The article also uncovers a new avenue for research in the field of urban science in Africa.

On the one hand, it is generally understood that UA typically requires land for cultivation [17,49]. Hence, it is important for the subject of UA to be adequately discussed in architecture and urban planning studies [50]. On the other hand, health has always been a fundamental consideration in early urban planning [51]. Studies clearly indicate that UA involves health issues and is an integral part of urban life [52]. However, this study demonstrates that urban planning experts do not show sufficient interest in UA as a mediator of urban health [53].

To address this weakness, several initiatives could be undertaken via urban planning specialists [54]. These include zoning UA within the urban fabric and providing infrastructure and networks, such as drinking water and electricity for watering plants, tasks that fall under the purview of urban planners and architects [55]. Additionally, regulations could offer incentives for integrating UA as closely as possible with inhabitants, promoting the use of soilless, rooftop, and wall-mounted cultivation innovations to optimize surface area [56]. Furthermore, efforts to combat the use of toxic products can be undertaken with the support of institutions responsible for agriculture and health [57].

*4.3. Establishment of an Overview of the Positive and Negative Health Impacts of UA in African Cities*

4.3.1. The Negative Health Impacts of UA Are Not Negligible

This study reveals that the topics that have been the most constantly discussed for over a decade are waste management support, wastewater irrigation risks, pesticides/insecticides, and contamination dangers, with the most activity occurring between 2007 and 2020. The finding might explain what danger UA represents for human health and indicates that a somewhat more attention should be given to topics under the global umbrella of UA and health [41]? This fact could be a salutary response to the rising injunction to grow food in cities. Though precautions should be taken to avoid making UA a cause of urban health degradation [58].

However, if research continues this negative note, would UA lead to fewer urban health problems? The subject is especially relevant if the city is planned in conjunction with UA. Holistically, considering institutional, policy, and spatial planning aspects, there is a need to provide tools to raise awareness among urban farmers, provide subsidies for access to cleaner and safer plant health products, preserve UA land in the healthiest possible locations within the city, and provide clean water [59].

4.3.2. Researchers Are Looking at UA from a Dual Perspective in Terms of Its Health Impacts

Based on these data illustrating the study contexts of the subject, it appears that UA has both positive and negative impacts on human health in Africa, and in almost equal proportions [60]. The urban planning discipline still has room to study the subject of UA

and health in order to reveal the urban planning factors that mitigate the risks or enhance the benefits of UA in African cities.

Either way, sufficient documentation of two types of impact in the literature confirms that both elements are to be taken seriously. The result of the analysis is therefore indicative of a gap to be filled in terms of research on UA impacts.

### 4.4. Recommendations

The recommendations from this discussion are to initiate more master's theses and doctoral dissertations on the triptych of UA, health, and urban planning; for practitioners to work more with the subject in creating and using their tools; and to use mixed methods adapted to the socio-economic and cultural realities of Africa. Mixed methods may help researchers move towards a balance of methodological types in this kind of research.

In addition, taking advantage of the growing ubiquity of information and communication technologies and alternative channels of publication on the African continent, as demonstrated by [61], would be beneficial for experts in urban planning in Africa to become more interested in studying the topic of UA and health and, in so doing, give equal importance to the positive and negative impacts of research on UA and health. The choice of increasingly contemporary formats and publication channels, such as instructional visual content and social networks, could be more and more adapted to reality [62]. With the increasing disruption of technological tools, such as smartphones and their applications, in Africa, African scientists can seize these means to increase their visibility in general and publish about UA and health [63]. This would also allow them to reach as many of their target populations as possible.

Another recommendation is that many African countries would benefit from research on UA and health, meaning that the results must be as comprehensive as possible. Additionally, scientists from laboratories in African countries would be best placed to research the subject in Africa to obtain results that consider the realities of the people immersed.

Integrating innovative practices such as vertical farming, hydroponics, and aeroponics to minimize the risks of contamination from the immediate environment and respond to the growing land scarcity are other possible solutions [63]. Also, future studies on this topic could add value by adopting the mixed method. This approach would also promote multidisciplinarity, which is increasingly recommended in scientific research [64–66].

Moreover, it appears from the analysis that UA has a mitigated impact on health. Therefore, researching whether the integration of UA into urban planning tools like master plans alleviates risks or enhances the benefits of UA concerning the food security and health indicators of the people could be important and bring something new to scientific research. It would also be legitimate not to focus on only one type of impact but to continue to study them in parallel in order to determine how negative impacts can be mitigated and positive impacts enhanced [63].

### 4.5. VOSviewer Results Discussion

In summary, these visualizations provide researchers with a powerful tool to better understand the thematic structure and connections within each dataset. It helps identify critical areas of research interest, highlight potential interdisciplinary intersections, and guide further exploration and analysis of the data. The different clusters within each dataset can serve as a roadmap for researchers to delve deeper into specific topics of interest and uncover hidden insights within the data. The three results from the three databases have several common elements and themes.

All three results cluster terms based on their co-occurrence patterns in the respective datasets. This clustering helps identify thematic groups and relationships between terms. In each visualization, some prominent themes or clusters stand out, indicating areas of significant research interest or common topics within the datasets. These themes include topics like "malaria", "survey", "insecticide", "Ghana", and "diseases".

The visualizations show that the research topics often intersect and have connections across different domains. For example, the relationship between "diseases" and "policies" is evident in the Web of Science data, and the interaction between access, prevalence, and policies is highlighted in Figure 11 of the Scopus database. The connections provide insights into the strength of relationships between terms and how they are bibliographically coupled.

"Diseases" and geographical location are common themes in Web of Science and PubMed, suggesting that research in these datasets often revolves around issues related to health and geography.

Overall, the commonalities in these results include the use of visualizations to reveal thematic structures, interconnections, and prominent areas of research interest in the respective datasets. These visualizations offer valuable insights, including elements of the results obtained in the Excel and QGIS analyses, such as Ghana as the most cited case study and UA's positive and negative impacts on health.

*4.6. Limitations*

The typological, chronological, thematic, geographic, and methodological analyses in the research offer the possibility of hypothesizing about the characteristics of the literature on UA and health in Africa. It is important to identify biases that may have influenced the outcomes. The findings should therefore be considered in this context.

One area for improvement is that the articles were selected from the databases without being separated by metadata, such as their impact on publications and their citation metrics.

Also, as the analyses were performed manually and Google Scholar did not automatically provide the main fields of study of the author in the columns of the CSV file provided through Zotero, only Scopus, Web of Science, and PubMed data were used for the "main field of affiliation of authors" analysis and the methodological assessment. The "chronological evolution analysis of topics" was only based on Scopus alone and was the first-ranked on the list of databases.

In addition, only data from Scopus, Web of Science, and PubMed were used for certain analyses, to the exclusion of other databases. This exclusion may limit the completeness and representativity of the results, as important publications from other sources may have been ignored.

Another limitation is that the databases from which the data were extracted are mostly in English. Google Scholar offered articles in other languages, but the search form was elaborated in English using a simple and easy-to-replicate method in order to harmonize result especially since the search was also conducted in English. The results might have been different, for example, in terms of the countries of the study sites or the affiliations of the authors. Other research initiatives could be conducted in other languages or more comprehensively to compare results. A review could also consider other publication platforms to diversify the sources.

One limitation of this research is that the publications studied come from both the peer-reviewed and the gray literature. These mixed origins may mean that academic rigor was not applied equally to all publications. Also, the search equation was not adapted to each database. This could have produced a different result than would otherwise have been objected to. Another limitation is that only one researcher collected, extracted, analyzed, and interpreted the data. This may mean that some relevant studies have been omitted.

The fact that the study did not include the years between 2021 and 2023 in this research is also a limitation. This was caused by the author's commitment to the remaining PhD years.

Finally, this article was written several months after the data were collected, and with the evolution of artificial intelligence, for example, some data might have changed when applying the research equation, and interpretations might have been different.

## 5. Conclusions

This paper empirically analyzes the publications of UA studies in Africa related to the health theme. It assesses the extent to which they link to urban planning. The novel contribution of this study is to expose the need for the African architectural and urban planning discipline to take a more significant and a more concrete interest in the topic and incorporate it into studies, planning documents, and development projects. The paper discloses another avenue of research in African urban science: UA, health, and urban planning.

The primary message is that planners and architects should perform more research and experimentation in laboratories and firms. Practically, this would mean, for instance, studying how the study integrated UA into master plans, houses, building plans, and construction to contribute to more positive and less harmful UA health outcomes. Some international policy implications of the study's findings include the following points:

- The need to raise awareness of UA issues in Africa. The results underline the growing importance of UA for public health in Africa. This may encourage international bodies such as the World Health Organization (WHO) and the FAO, as well as foreign governments, to consider the specific needs and challenges of UA in Africa in their development policies and programs;
- The promotion of sustainable UA. The study's findings highlight the necessity of integrating UA into urban policies and planning. This may encourage African countries to develop strategies and regulations to support sustainable UA that contributes to food security, public health, and the reduction in greenhouse gas emissions;
- International collaboration must begin, as the results of the study suggest that UA is an area of growing interest. International collaborations in research, best-practice sharing, and technical assistance could help African countries take advantage of UA in ways that benefit health and the environment;
- There would also be a need to support training and education. For UA to be effectively integrated into policy and practice, it is essential to train urban planners, architects, and public health professionals. International organizations could support training and education in these fields in Africa;
- New urban health policies must be developed. The results of the study highlight the essential role of UA in the public health of African cities. This could encourage international organizations to promote the development of urban health policies that integrate UA as a key element of health promotion. In summary, the international policy implications of this study's findings concern the recognition of the importance of UA in Africa for public health and the environment, as well as the promotion of policies and practices to support sustainable UA that benefits the citizens of African cities.

Possible avenues for future studies include replicating the study with other types of literature reviews, other databases, larger time scopes, and even longitudinal studies. Additional studies could consider the other continents as well. Future research could perform in-depth analysis on the effects of UA on the prevalence of non-communicable diseases, on the food security of urban populations, and on reducing the risk of diseases linked to air and water quality. It would be interesting to conduct systematic evaluations of UA-related policies and interventions in Africa to determine their effectiveness in terms of public health, food security, and the environment.

This could help identify best practices and lessons to be learned. Further research could be undertaken to understand how architects and planners can play a more active role in promoting UA and urban health in Africa. This could include surveys of planners' knowledge, skills, and attitudes towards UA. Comparative studies between African cities and other regions of the world could be carried out to assess similarities and differences in the impact of UA on urban health. This would enable transferable lessons to be learned.

The implementation of these recommendations from this study will help to provide a more legitimate framework for UA in the African urban landscape in order to reduce the health risks it poses, while increasing the likelihood that the positive impacts of UA

on urban health will be optimal for the well-being of all city dwellers and, if possible, the entire territory. Hence, urban experts can contribute through UA planning to keeping the African city and the world city, in general, as healthy as possible. In doing so, they will have helped cities to breathe, to eat, to be better, and, quite simply, to live.

**Author Contributions:** Conceptualization, A.A.K. and J.C.; methodology, A.A.K. and A.F.K.M.; software, A.A.K.; validation, J.C., B.J.-C.M. and A.F.K.M.; formal analysis, A.A.K., A.F.K.M., B.J.-C.M. and J.C.; investigation, A.A.K.; resources, J.C.; data curation, A.A.K.; writing—original draft preparation, A.A.K.; writing—review and editing, A.A.K., A.F.K.M., B.J.-C.M. and J.C.; visualization, A.A.K.; supervision, J.C., B.J.-C.M. and A.F.K.M.; project administration, J.C.; funding acquisition, J.C. All authors have read and agreed to the published version of the manuscript.

**Funding:** This work was totally supported by the Swiss National Science Foundation (SNF#183577) Sinergia Project—African Contribution to Global Health: Circulating Knowledge and Innovations (https://www.globalhealthafrica.ch/, accessed on 14 November 2023).

**Data Availability Statement:** The data presented in this study are openly available on Zenodo at https://zenodo.org/record/8043945, accessed on 14 November 2023, reference number 10.5281/zenodo.8043945.

**Acknowledgments:** The authors like to show their gratitude to Vitor Pessoa Colombo, Remi Jaligot, Marti Bosch, Ximena Salgado Uribe, Pablo Txomin Harpo de Roulet, Salifou Ndam, Gladys Ninoles, and Carine Micheloud from the CEAT team; Afriyie, Jürg Utzinger, Doris Osei, and med. Ipyn Eric Newbie from the Swiss TPH team; and Julia Tischler, Tanja Hammel, and Danelle van Zyl-Hermann from the University of Basel for guiding and supporting them throughout the data collection, writing, and review process.

**Conflicts of Interest:** The authors declare no conflict of interest.

**Data Citation:** Konou: Akuto Akpedze, Kemajou Mbianda, Armel Firmin, Baraka Jean-Claude Munyaka & Chenal, Jérôme. (2023). Urban Agriculture and Health in Africa. A Review. [Data set]. Zenodo. https://doi.org/10.5281/zenodo.8043945.

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
