# Peer review of "Two Decades of Architects’ and Urban Planners’ Contribution to Urban Agriculture and Health Research in Africa"

_urbansci, doi:10.3390/urbansci7040117_

Round 1

Reviewer 1 Report

Comments and Suggestions for Authors

This study investigated the relationship between urban agriculture and health in Africa. The results showed the evidence on urban agriculture is increasing from public health researchers, but not in the field of urban planning or architecture. In sum, this study engages with a subject of significant relevance both in terms of research and practical implications. Nevertheless, certain areas in the manuscript warrant refinement and elucidation. 

Comments on the Quality of English Language

This study investigated the relationship between urban agriculture and health in Africa. The results showed the evidence on urban agriculture is increasing from public health researchers, but not in the field of urban planning or architecture. In sum, this study engages with a subject of significant relevance both in terms of research and practical implications. Nevertheless, certain areas in the manuscript warrant refinement and elucidation. 

Author Response

                                                                                Akuto Akpedze Konou

PhD Candidate at École Polytechnique Fédérale de Lausanne (EPFL)
Phone: +41 77 510 64 52
Email: [email protected]

Reviewer 1

Urban Science

                                                     6th October 2023

Dear Reviewer 1,

New title: Reviewing two decades of literature on architects’ and urban planners' contribution to urban agriculture and health research in Africa (2000-2020)

Thank you for considering our paper submission and to have reviewed it.

Please find attached our round 1 reviewed manuscript

We received excellent suggestions from all the reviewers and took them on board. Please find on the following pages the responses to the reviewers.

We hope that the modifications improve the value of our manuscript and are open to any new suggestions.

Thank you for your time in considering this submission. We look forward to hearing from you.

Yours sincerely,

Akuto Akpedze Konou

For research article

Response to Reviewer 1 Comments

1. Summary

Thank you very much for taking the time to review this manuscript. Please find the detailed responses below and the corresponding revisions highlighted in the re-submitted files.

2. Questions for General Evaluation

Reviewer’s Evaluation

Response and Revisions

Does the introduction provide sufficient background and include all relevant references?

Can be improved

Thank you for pointing this out. We agree with this comment. Therefore, we have improved it and specified the details in the point-by-point response table below.

Are all the cited references relevant to the research?

Can be improved

Thank you for pointing this out. We agree with this comment. Therefore, we have improved it and specified the details in the point-by-point response table below.

Is the research design appropriate?

Can be improved

Thank you for pointing this out. We agree with this comment. Therefore, we have improved it and specified the details in the point-by-point response table below.

Are the methods adequately described?

Can be improved

Thank you for pointing this out. We agree with this comment. Therefore, we have improved it and specified the details in the point-by-point response table below.

Are the results clearly presented?

Can be improved

Thank you for pointing this out. We agree with this comment. Therefore, we have improved it and specified the details in the point-by-point response table below.

Are the conclusions supported by the results?

Can be improved

Thank you for pointing this out. We agree with this comment. Therefore, we have improved it and specified the details in the point-by-point response table below.

3. Point-by-point response to Comments and Suggestions for Authors

Comments 1: Title should be revised so that it can highlight the purpose of the study well.

Response 1: Thank you for pointing this out. We agree with this comment. Therefore, we have changed the title to: "Reviewing two decades of literature on architects' and urban planners' contribution to urban agriculture and health research in Africa (2000-2020)" – page 1, lines 2-4.

Comments 2: Authors should refer to the Authors' guidelines and prepare the manuscript according to the journal’s layout.

Response 2: Thank you for pointing this out. We agree with this comment. Therefore, we have changed all the titles under the Urban Science layout.

Comments 3: The English grammar and style should be checked throughout the manuscript.

Response 3: Thank you for pointing this out. We agree with this comment. Therefore, we have used MDPI's English editing department services to check and correct the English of this second manuscript version before resubmitting it.

Comments 4: The authors should avoid using pronouns such as “we”, “our” and “us” in the whole text (e.g., line 116 and 120).

Response 4: Thank you for pointing this out. We agree with this comment. Therefore, we have changed the “we” phrases into passive voice: “Microsoft Excel 16 was used” and “a typological profile of these publications was established” – page 3, lines 116, 120.

Comments 5: The Abstract needs improvement in terms of conveying the main research problem, analyses, and key results more explicitly. Currently, the statement of results and conclusions in the abstract lacks the necessary details and specific data that are essential to understand the study's significance.

Response 5: Thank you for pointing this out. We agree with this comment. Therefore, we have introduced the results with the phrase “The results displayed” on the line 20 and added the missing results, to add more clarity about the results, and we have reformulated and restructured – page 1, lines 11-24.

Comments 6: Furthermore, the Abstract should explicitly indicate the research's originality and highlight its contribution to the international literature. It is imperative to emphasize how the study introduces novel insights and advances the existing knowledge in the field.

Response 6: Thank you for pointing this out. We agree with this comment. Therefore, we have added new sentences to show the originality of the research and the contribution to international literature – page 1, lines 23, 24.

Comments 7: The authors should avoid using keywords that already mentioned in the title (e.g., Africa and urban agriculture) and replace them with new relevant words in the text.

Response 7: Thank you for pointing this out. We agree with this comment. Therefore, we have, throughout the text, transformed specific terms into « Africa » by « African continent » or « African territory » where it was possible to do so. We have also changed the word "Agriculture" to farming for example, on lines 34 of page 1.

Comments 8: The Introduction provides an overview of urban agriculture (UA) and its historical context in Africa. However, it could benefit from a clearer structure, with distinct paragraphs for different aspects, such as historical background, contemporary relevance, and the research objectives.

Response 8: Thank you for pointing this out. We agree with this comment. Therefore, we have restructured the introduction into 4 different parts: historical background, contemporary relevance, Problematic and gaps, and the research objectives – pages 1 and 2, lines 28, 42, 52, and 84.

Comments 9: The Introduction mentions the research objectives but doesn't clearly outline what specific questions or issues the study aims to address. Providing a concise and precise statement of the research objectives would help set the stage for the rest of the paper.

Response 9: Thank you for pointing this out. We agree with this comment. Therefore, we have restructured the introduction, added a paragraph about the problematic, and distinguished each objective with dashes. – page 2, lines 97-103.

Comments 10: The significance of the study should be added in the Introduction. The Introduced problems are all relevant and interesting, but it is not clear what the exact research problem for the paper is. The authors should be able to highlight the need for the study. Why is it timeliness to explore such a study? What makes this study different from other studies in the world? Are the findings different from prior academic studies that were conducted elsewhere, if any?

Response 10: Thank you for pointing this out. We agree with this comment. Therefore, we have added the 3 points to the article: Why is it timeliness to explore such a study? What makes this study different from other studies in the world? We have also stated how the findings are different from prior academic studies that were conducted elsewhere, if any? – page 2, lines 70, 75, 81,82.

Comments 11: Page 2, lines 70-71: This sentence “Africa being known as continent with an unprecedent growing urban population” could be rephrased for greater clarity.

Response 11: Thank you for pointing this out. We agree with this comment. Therefore, we have removed the sentence.

Comments 12: Page 2, lines 52-89: The term of “Urban Agriculture” should be revised to “UA”. It should be also checked and revised in the rest of the manuscript.

Response 12: Thank you for pointing this out. We agree with this comment. Therefore, we have revised “Urban Agriculture”to “UA” throughout the paper.

Comments 13: Page 2, line 78: This sentence “Some authors have also been discussing the integration of urban agriculture into…” is vague. Please revise the sentence and state your idea more clearly.

Response 13: Thank you for pointing this out. We agree with this comment. Therefore, we have removed the sentence.

Comments 14: It's crucial to specify the sources from which you collected data (e.g., Scopus, PubMed, Web of Sciences, Google Scholar) right at the beginning of Methodology section.

Response 14: Thank you for pointing this out. We agree with this comment. Therefore, we have specified right at the beginning of Materials and Methods section the sources from which we collected data – page 3 and lines 106, 107.

Comments 15: The query string used for data collection is provided, which is helpful. However, there's a minor typo in the query string. It should be ("urban agriculture" OR "urban farming") AND Africa AND ("health" OR "global health" OR "Urban health") with a closing parenthesis at the end.

Response 15: Thank you for pointing this out. We agree with this comment. Therefore, we have revised the query string to: ("urban agriculture" OR "urban farming") AND Africa AND ("health" OR "global health" OR "Urban health") with a closing parenthesis at the end – page 3, lines 112-114.

Comments 16: While you mention the “Data cleaning process”, you could elaborate a bit more on the criteria used for cleaning and any challenges encountered. This would provide transparency about data quality.

Response 16: Thank you for pointing this out. We agree with this comment. Therefore, we have added a paragraph giving the information about data cleaning - page 4, line 127-131.

Comments 17: Page 3, line 123: The sentence of "The results obtained from this analysis informed the preferential types of publications by field…" should be revised and restructured.

Response 17: Thank you for pointing this out. We agree with this comment. Therefore, we have revised and restructured the sentence into: “The results of this analysis provided insights into the preferred publication types within each field.” – page 3, line 141, 142

Comments 18: Page 4, line 136: This sentence “Thematic analysis was, therefore, the place to evaluate these…” need to be revised.

Response 18: Thank you for pointing this out. We agree with this comment. Therefore, we have rephrased the sentence to “Thematic analysis served as the appropriate method for assessing these diverse subjects.” – page 4, line 153, 154.

Comments 19: While you provide data on the number of publications and trends, the interpretation of these findings could be more explicit. Explain the significance of certain trends or findings and their implications for your research objectives.

Response 19: Thank you for pointing this out. We agree with this comment. Therefore, we have added to the discussion, “The result emphasizes the objective of the urge for architects and urban planners to join the movement” – page 14, and line 331, 332.

Comments 20: In Figure 5, there's an inconsistency in the labeling of the horizontal axis. The years are labeled from 1999 to 2021, but the text mentions the analysis covering 2000 to 2020. Make sure the labels and the actual data align.

Response 20: Thank you for pointing this out. We agree with this comment. Therefore, we have corrected the horizontal axis labeling into 2000 to 2020 – page 5, Figure 2, lines 184.

Comments 21: All figures and tables should be cited in the main text.

Response 21: Thank you for pointing this out. We agree with this comment. Therefore, we have checked and cited all the figures and tables.

Comments 22: The discussion section is well structured, just please revise the head of section “Discussion of results” to the “Discussion”.

Response 22: Thank you for pointing this out. We agree with this comment. Therefore, we have revised the head of section “Discussion of results” to the “Discussion” – page 12, line 302.

Comments 23: The Conclusion section should effectively communicate concise and innovative information to the readers. Currently, it may be difficult to discern the novel contributions of this study. It is essential to provide a clear take-home message that emphasizes the main findings and their practical implications. Furthermore, in the Conclusion section, the focus should be on restating the main results and demonstrating how the research questions have been thoroughly examined and explained.

Response 23: Thank you for pointing this out. We agree with this comment. Therefore, we have restructured and enriched the conclusion following the points:

-        restating the main results and demonstrating how the research questions have been thoroughly examined and explained

-        the novel contributions of the study.

-        the clear take-home message that emphasizes the main findings and their practical implications.

– pages 18, 19, lines 572-579, 580, 581, 583-629

Comments 24: To enhance the section, it is important to enrich it with paragraphs discussing the international policy implications of the study's findings. This will provide valuable insights into how the research outcomes can inform and influence policy decisions at an international level.

Response 24: Thank you for pointing this out. We agree with this comment. Therefore, we have added paragraphs to discuss the international policy implications of the study's findings – pages 18, 19, lines 590-615.

Comments 25: In Conclusion, it is important to emphasize the novelty of the study, which distinguishes it from previous research. Additionally, it is crucial to discuss both the theoretical and practical implications of the study's findings. This includes exploring how the research contributes to the existing theoretical framework and its potential applications in real-world settings. Moreover, it is recommended to outline possible avenues for future studies that can build upon the current research and address any remaining gaps or unanswered questions.

Response 25: Thank you for pointing this out. We agree with this comment. Therefore, we have addressed the theoretical and practical implications of the study's findings in the two previous comments responses, and we have added possible avenues for future studies – page 19, lines 616-629.

4. Response to Comments on the Quality of English Language

Point 1: The English grammar and style should be checked throughout the manuscript.

Response 1: Thank you for pointing this out. We agree with this comment. Therefore, we have used MDPI's English editing department services to check and correct the English of this second manuscript version before resubmitting it.

5. Additional clarifications

None.

 Please also see the attachment.

Reviewer 2 Report

Comments and Suggestions for Authors

From the title of this manuscript, it is important that authors modify the title by specifying the kind of review. From my observation, it is more like a bibliometric analysis-type of review though it didn't follow the guidelines for a review of this kind.

Please, kindly rewrite the abstract section, salient findings were conspicuously missing.

The introduction was not well articulated. In my own opinion, the urban agriculture should be linked directly to food and nutrition security while health outcomes should be generated from it and not the opposite.

The background information for this study was not well presented. The knowledge gap(s) was not well articulated. 

From the methods, why was the search terminated in December 2020? I believe incredible papers can be extracted within 2021-2023 that can improve the results of this study. 

In line 74, what's the meaning of "as le"? What do you mean by "4portals outcome synthesised" in fig.1? The final number of included publications is missing in fig. 1. 

In line 157-159, what's the number of publications from Google scholar from your search procedures? 

In line 173-174, why the affiliations of authors from Scopus database excluded? What is the total number of extracted publications in this stage?

In line 198, why the inconsistency in reporting your results? If you know some databases may not provide suitable or usable results, you may stick to fewer databases or just only one reliable database usually Scopus or web of science. 

In line 259-267, you excluded papers from Google scholar database again. 

This review did not follow the basic guidelines of a 

bibliometric review. You can not use excel to create your mapping that will suit this kind of review. Use software like VOSviewer or Bibliometrix using R software. There are so many others but the two are commonly used.

It is highly recommended that authors use any of these software for this study. The real bibliographical mappings are not generated in Excel. 

Thank you.

Comments on the Quality of English Language

Needs some basic language editing and formating of figures.

Author Response

                                                                                Akuto Akpedze Konou

PhD Candidate at École Polytechnique Fédérale de Lausanne (EPFL)
Phone: +41 77 510 64 52
Email: [email protected]

Reviewer 2

Urban Science

                                                     6th October 2023

Dear Reviewer 2,

New title: Reviewing two decades of literature on architects’ and urban planners' contribution to urban agriculture and health research in Africa (2000-2020)

Thank you for considering our paper submission and to have reviewed it.

Please find attached our round 1 reviewed manuscript

We received excellent suggestions from all the reviewers and took them on board. Please find on the following pages the responses to the reviewers.

We hope that the modifications improve the value of our manuscript and are open to any new suggestions.

Thank you for your time in considering this submission. We look forward to hearing from you.

Yours sincerely,

Akuto Akpedze Konou

For review article

Response to Reviewer 2 Comments

1. Summary

Thank you very much for taking the time to review this manuscript. Please find the detailed responses below and the corresponding revisions highlighted in the re-submitted files.

2. Questions for General Evaluation

Reviewer’s Evaluation

Response and Revisions

Does the introduction provide sufficient background and include all relevant references?

Must be improved

Thank you for pointing this out. We agree with this comment. Therefore, we have improved it and specified the details in the point-by-point response table below.

Are all the cited references relevant to the research?

Must be improved

Thank you for pointing this out. We agree with this comment. Therefore, we have improved it and specified the details in the point-by-point response table below.

Is the research design appropriate?

Must be improved

Thank you for pointing this out. We agree with this comment. Therefore, we have improved it and specified the details in the point-by-point response table below.

Are the methods adequately described?

Must be improved

Thank you for pointing this out. We agree with this comment. Therefore, we have improved it and specified the details in the point-by-point response table below.

Are the results clearly presented?

Must be improved

Thank you for pointing this out. We agree with this comment. Therefore, we have improved it and specified the details in the point-by-point response table below.

Are the conclusions supported by the results?

Must be improved

Thank you for pointing this out. We agree with this comment. Therefore, we have improved it and specified the details in the point-by-point response table below.

3. Point-by-point response to Comments and Suggestions for Authors

Comments 1: From the title of this manuscript, it is important that authors modify the title by specifying the kind of review. From my observation, it is more like a bibliometric analysis-type of review though it didn't follow the guidelines for a review of this kind.

Response 1: Thank you for pointing this out. We agree with this comment. Therefore, we have changed the title to: "Reviewing two decades of literature on architects' and urban planners' contribution to urban agriculture and health research in Africa (2000-2020)” – page 1, and line 2-4.

Comments 2: Please, kindly rewrite the abstract section, salient findings were conspicuously missing.

Response 2: Thank you for pointing this out. We agree with this comment. Therefore, we have revised the abstract adding the missing results – page 1, lines 11-24.

Comments 3: The introduction was not well articulated. In my own opinion, the urban agriculture should be linked directly to food and nutrition security while health outcomes should be generated from it and not the opposite.

Response 3: Thank you for pointing this out. We agree with this comment. Therefore, we have restructured the introduction so that urban agriculture is directly linked to food and nutrition security while health outcomes are generated:

-        Paragraph 1 talks about UA generalities

-        Paragraph 2 talks about UA and food security

-        Paragraph 3 talks about UA and Public Health

(Page 1,2, Lines 29-62)

Comments 4: The background information for this study was not well presented. The knowledge gap(s) was not well articulated.

Response 4: Thank you for pointing this out. We agree with this comment. Therefore, we have restructured the introduction into 4 different parts: historical background, contemporary relevance, Problematic and gaps, and the research objectives – pages 1 and 2, lines 28, 42, 52, and 84.

We have also restructured the introduction, added a paragraph about the problematic, and distinguished each objective with dashes. – page 2, lines 97-103.

Finally, we have added the 3 points to the article: Why is it timeliness to explore such a study? What makes this study different from other studies in the world? We have also stated how the findings are different from prior academic studies that were conducted elsewhere, if any? – page 2, lines 70, 75, 81, 82.

Comments 5: From the methods, why was the search terminated in December 2020? I believe incredible papers can be extracted within 2021-2023 that can improve the results of this study.

Response 5: Thank you for pointing this out. We agree with this comment. Therefore, we would like to explain that the data collection of this research was done until December 2020, then Ph.D. field work trips were carried out by the principal author, and then data analysis was carried out in 2022, and submission was possible in 2023.

Comments 6: In line 74, what's the meaning of "as le"? What do you mean by "4portals outcome synthesised" in fig.1? The final number of included publications is missing in fig. 1.

Response 6: Thank you for pointing this out. We agree with this comment. Therefore, we have removed the "as le" and the title “"4portals outcome synthesised" and added the final number of included publications – page 3, line 125.

Comments 7: In line 157-159, what's the number of publications from Google scholar from your search procedures?

Response 7: Thank you for pointing this out. We agree with this comment. Therefore, we have added that the number of Google Scholar publications gathered is n=150 publications – page 5, line 177.

Comments 8: In line 173-174, why the affiliations of authors from Scopus database excluded? What is the total number of extracted publications in this stage?

Response 8: Thank you for pointing this out. We agree with this comment. Therefore, we would like to specify that we have excluded Scopus in some studies because it was tedious work that one person could not do manually according to the number of publications. The total at this stage was 390 publications. Thank you again.

Comments 9: In line 198, why the inconsistency in reporting your results? If you know some databases may not provide suitable or usable results, you may stick to fewer databases or just only one reliable database usually Scopus or web of science.

Response 9: Thank you for pointing this out. We agree with this comment. Therefore, like in the previous comment, we would like to specify that we have excluded Scopus in some studies because it was tedious work that one person could not do manually according to the number of publications. The total at this stage was 390 publications. Thank you again.

Comments 10: In line 259-267, you excluded papers from Google scholar database again.

Response 10: Thank you for pointing this out. We agree with this comment. Therefore, like in the previous comment, we would like to specify that we have excluded Scopus in some studies because it was tedious work that one person could not do manually according to the number of publications. The total at this stage was 390 publications. Thank you again.

Comments 11: This review did not follow the basic guidelines of a bibliometric review. You can not use excel to create your mapping that will suit this kind of review. Use software like VOSviewer or Bibliometrix using R software. There are so many others but the two are commonly used. It is highly recommended that authors use any of these software for this study. The real bibliographical mappings are not generated in Excel.

Response 11: Thank you for pointing this out. We agree with this comment. Therefore, we have precised that the maps are obtained using QGIS 3.32.2 – page 4, line 135, and page 11, line 271, 279.

4. Response to Comments on the Quality of English Language

Point 1: Needs some basic language editing and formatting of figures.

Response 1: Thank you for pointing this out. We agree with this comment. Therefore, we have used MDPI's English editing department services to check and correct the English of this second manuscript version before resubmitting it.

5. Additional clarifications

None.

Please also see the attachment.

Thank you.

Round 2

Reviewer 1 Report

Comments and Suggestions for Authors

The authors have successfully addressed the comments. Nonetheless, there are some minor issues that require their attention.

Comments on the Quality of English Language

The moderate English revision is required.

Author Response

Akuto Akpedze Konou

PhD Candidate at École Polytechnique Fédérale de Lausanne (EPFL)
Phone: +41 77 510 64 52
Email: [email protected]

Reviewer 1

Urban Science

19th October 2023

Dear Reviewer 1,

New title: Two decades of architects’ and urban planners’ contribution to urban agriculture and heath research in Africa

Thank you for considering our paper submission and to have reviewed it.

Please find attached our round 2 reviewed manuscript

We received excellent suggestions from you and took them on board. Please find on the following pages the responses to your comments.

We hope that the modifications improve the value of our manuscript, and we are open to any new suggestions.

Thank you for your time in considering this submission. We look forward to hearing from you.

Yours sincerely,

Akuto Akpedze Konou

Reviewer 2 Report

Comments and Suggestions for Authors

Thank you for submitting the revised version of this manuscript. However, there is a need to still have a second look at the title and modify it. I suggest removing the word "reviewing" and "literature"from the title. "Two decades of architects' and urban planners" contribution to urban agriculture and heath research in Africa". You may not need to add 2000-2020 since you have started the title with "two decades".

Since, authors prefer not to employ the conventional bibliographical mappings using VOSviewer or Bibliometrix or any other software, it is important that you give reasons (rationale for this departure) for this. 

Please, kindly reduce the conclusion section. Let it be a summary of you work alone and not repeating your findings.

Some of the suggestions/comments made on the original version were not attended to satisfactorily. I employ authors to attend to them. Thank you.

Comments on the Quality of English Language

Fine

Author Response

Akuto Akpedze Konou

PhD Candidate at École Polytechnique Fédérale de Lausanne (EPFL)
Phone: +41 77 510 64 52
Email: [email protected]

Reviewer 2

Urban Science

19th October 2023

Dear Reviewer 2,

New title: Two decades of architects’ and urban planners’ contribution to urban agriculture and heath research in Africa

Thank you for considering our paper submission and to have reviewed it.

Please find attached our round 2 reviewed manuscript

We received excellent suggestions from you and took them on board. Please find on the following pages the responses to your comments.

We hope that the modifications improve the value of our manuscript, and we are open to any new suggestions.

Thank you for your time in considering this submission. We look forward to hearing from you.

Yours sincerely,

Akuto Akpedze Konou

Round 3

Reviewer 2 Report

Comments and Suggestions for Authors

Fine

Comments on the Quality of English Language

The editor can make final publication decision on this revised version. Thank you.